# Does 8-Week Resistance Training with Slow Movement Cadenced by Pilates Breathing Affect Muscle Strength and Balance of Older Adults? An Age-Matched Controlled Trial

**DOI:** 10.3390/ijerph191710849

**Published:** 2022-08-31

**Authors:** Ilha G. Fernandes, Maria C. G. S. Macedo, Matheus A. Souza, Gabriela Silveira-Nunes, Michelle C. S. A. Barbosa, Andreia C. C. Queiroz, Edgar R. Vieira, Alexandre C. Barbosa

**Affiliations:** 1Postgraduate Program in Physical Education, Federal University of Juiz de Fora, Juiz de Fora 35020-360, MG, Brazil; 2Musculoskeletal Research Group (NIME), Department of Physical Therapy, Federal University of Juiz de Fora, Juiz de Fora 35020-360, MG, Brazil; 3Department of Physical Education, Federal University of Juiz de Fora, Juiz de Fora 35020-360, MG, Brazil; 4Department of Physical Therapy, Florida International University, Miami, FL 33199, USA

**Keywords:** physical therapy, training, breathing, postural balance, muscle strength, aging

## Abstract

This study investigated the balance and dorsiflexion strength of older adults after eight weeks of resistance training, with the exercise velocity cadenced by the Pilates breathing technique and the volume modulated by the session duration. Forty-four older adults were divided into two groups: resistance training (TR; n = 22) and resistance training with the Pilates breathing technique cadencing all exercises (TR + P; n = 22), both during eight weeks. The total exercising volume was controlled by time of execution (50 min/session). The dorsiflexion strength and balance were assessed. The RT group showed higher dorsiflexion strength after the protocol: Right (RT = 29.1 ± 7.7 vs. RT + P = 22.9 ± 5.2, *p* = 0.001) and Left (RT = 29.5 ± 6.9 vs. RT + P = 24.0 ± 5.2, *p* = 0.001). All balance parameters were improved in RT + P group compared to its own baseline: Path Length (cm) (pre = 71.0 ± 14.3 vs. post = 59.7 ± 14.3, *p* = 0.003); Sway Velocity (cm/s) (pre = 3.6 ± 0.7; post = 2.9 ± 0.7; *p* = 0.001); Sway Area (cm^2^) (pre = 8.9 ± 5.3 vs. post = 5.7 ± 2.1, *p* = 0.003); Excursion Medio Lateral (cm) (pre = 3.0 ± 0.7 vs. post = 2.6 ± 0.5 cm, *p* = 0.002); and Excursion AP (cm) (pre = 3.6 ± 1.4 vs. post = 2.8 ± 0.7 cm, *p* = 0.010). Resistance training using slower velocity movement cadenced by Pilates breathing technique produced balance improvements compared to baseline (moderate to large effect sizes), but no between-group effect was observed at the end of the protocol. The dorsiflexion strength was higher in the RT group compared to RT + P group.

## 1. Introduction

Progressive worldwide growth of the older population is a well-established trend [1,2]. Unfortunately, aging is often associated with losses in neuromuscular function and physical performance [3,4] which may lead to an increased risk of falling and fractures [3,5,6]. There was a 200% increase in the mortality rate from falls in the elderly between 1996 and 2012 in Brazil [7]. Femur fractures are the fractures that most require hospitalization, carrying a high risk of mortality and great loss of independence [8]. Muscle weakness is a key factor that determines postural control and faster recovery responses after sudden losses of balance in order to prevent falls [3,9,10]. The loss of muscle mass leads to declines in muscle force output, mainly due to decreases in voluntary muscle activation, reduced number of motor units and lower size of muscle fibers [3,9,10]. Poor balance has been cited as an important factor associated with physical decline with aging [10,11]. Critical markers in balance control were correlated with an increased risk of falls [12,13]. Compared with young adults, older people show larger center-of-pressure displacements and sway velocity in semi-static stance under different conditions (e.g., eyes opened/closed; stable/unstable surface [11,13].

In order to avoid raising the incidence of fractures in the coming decades, effective prevention programs need to be implemented [14]. Exercising has been proposed as a successful strategy to mitigate those risks by improving physical capacities, quality of life, and frailty among older people, counteracting lower-limb weakness, poor balance, and the physical impairments induced by injurious falls [4,15,16]. Benefits on functional parameters of older people were yielded by training interventions, such as resistance, balance, endurance, coordination, and multi-modal exercises (i.e., the combination of strength, endurance, and balance) [17,18]. Many types of exercising methods have been studied as a viable and easy-to-perform way to prevent the quantifiable risks in terms of physical parameters [18,19,20].

Among the various types of exercises, resistance training (RT) has been able to mitigate the impact of aging and muscle disuse [21], showing a moderate positive effect on elderly disability [22]. This type of exercise requires the muscles to contract against a force, which can be offered by rubber bands, free weights, machines, other accessories, or the body weight itself [23]. RT has been shown to be effective in minimizing and even reversing functional losses, helping maintain muscle mass and improve muscle strength [24,25,26]. Its effects on postural balance are still open to discussion. Some studies show its positive effects [27] and others do not find any favorable result [28].

The Pilates method (or simply Pilates) combines coordinated breathing, concentration, technical accuracy, smooth movement transitions, controlled posture and movements, and the tightening of the deep abdominal, lumbar multifidus, and pelvic floor muscles for stabilization during the exercises [19,29,30]. Pilates has been found to increase strength and flexibility [31], and balance [32,33,34,35,36]. The most frequently reported principle of the Pilates is the breathing technique, suggesting that breathing is critical to coordinate the other principles, providing cadence to the exercise and regulating the level of muscle contraction [30,37]. In fact, while performing the same exercise, previous studies showed higher levels of muscle activity and force output when the Pilates breathing technique was performed [29,37]. These findings allow us to hypothesize that RT associated with Pilates breathing will promote greater stability and improve balance.

The RT cadence by Pilates breathing makes the exercise velocity execution slower. Usually, the exercise velocity is prescribed by the time needed to perform a concentric and eccentric phase [38]. A study modulated exercise velocity in young people and the results indicated that low load and slow movement training is as effective as normal training for strength and muscle mass gains [39]. Thus, low load and slow movement started to be investigated in older people. Evidence suggests that slow movement with moderate load [40] and even low load [41] can be an effective method to gain muscle mass and strength in older people.

Based on those findings, reducing the RT movement velocity using the Pilates breathing method may increase lower limb muscle recruitment, promoting strength and balance improvements. However, the studies that investigate how exercise velocity affects strength and hypertrophy controlled exercise performance time, but kept the same number of sets and repetitions in different groups [42]. Thus, the session time was often longer for those who performed the exercise slower. It is unknown whether similar results would be observed if the training volume were modulated by the duration of the session and not only by the number of sets and repetitions. The present study assessed the postural balance and the dorsiflexion strength of older adults after eight weeks of RT with exercise velocity cadenced by Pilates breathing technique, while the exercise volume was modulated by the session duration.

## 2. Materials and Methods

### 2.1. Participants

This is an age-matched controlled trial. The participants were recruited by public invitation through folders and personal contacts. Recruitment began in February 2018. In this moment, it was identified whether the volunteers met the inclusion criteria. A convenience sample of 81 community-dwelling older adults presented for the study. A preliminary assessment was performed to identify whether the volunteers met the inclusion criteria. The data was collected during the afternoon (1:00–6:00 PM). Prior to testing, the participants were familiarized with all physical assessment procedures. The sample had to be physically independent (level 3 or 4 on Functional Status; age ≥ 60 years old) [43], and cognitively able to understand the procedures (score > 21 on the Mini-Mental State Examination for people with low education) [44]. Exclusion criteria was as follows: cardiovascular disease, unstable proliferative retinopathy, end-stage renal disease, and uncontrolled hypertension [45]. All participants were free of any knee or hip injury which could affect their balance. The participants were also cleared from medication that could cause dizziness as a side effect. To assess the interaction among the usual aging variables that could interfere with balance assessments, all participants reported their history of falls in the previous 12 months. The international physical activity questionnaire was used to classify the volunteers’ physical activity level [46]. The Brazilian version of the Falls Efficacy Scale (FES-I-Brazil) was used to determine the participants’ level of fear of falls [47]. They were also previously medically tested for diabetes using the fasting blood sugar test (diabetic > 7 mmol/L and non-diabetic < 5.6 mmol/L). A calibrated hydraulic hand dynamometer (Saehan Co., Changwon, South Korea) was used for right handgrip strength measurement in baseline. The patient performed the test while sitting with their arms along the body, with the elbow joint in 90° flexion and the forearm and the wrist in a neutral position. The participant maximally gripped the handle of the dynamometer for up to 3 seconds, three times with the right hand (~10 s of rest), then the average value (in kilograms-force) was calculated.

The screening continued until 44 volunteers were included in the study, according to the sample size calculation. The a priori sample size calculation was based on a previous study [48], considering the effect size of 1.22, the alpha level of 5%, and a 95% power, returning a total of 32 individuals. Taking into account a sample loss of 30%, 44 volunteers were selected (22 per experimental group). After the first screening, 37 subjects were excluded, and 44 subjects followed for evaluation of balance and dorsiflexor strength. Interventions started in July 2018, right after balance and dorsiflexor strength assessments. They were equally divided into 2 groups: RT alone (RT group; n = 22) and RT with Pilates breathing technique (RT + P group; n = 22). The group’s assignment was performed following an age-matched order to balance both groups at baseline. 

The study was conducted in accordance with the Declaration of Helsinki, and the protocol was approved by the Ethics Committee of Federal University of Juiz de Fora (Project identification code 63775316.1.0000.5147), Ethics Board approval number 2.001.171, the study can be consulted at https://plataformabrasil.saude.gov.br/. All participants were informed about the benefits and risks involved and signed an informed consent form prior to participation. This study was registered in the Brazilian clinical trials registry (protocol number RBE-84GG5W). No practice at home was requested. Attendance was, therefore, taken as compliance with the assessment protocol. No co-interventions were performed in either group, and no adverse effects were reported by any participant during any procedure.

### 2.2. Balance Assessment

The BTrackS Balance Plate (Balance Tracking System, San Diego, CA, USA) was used to assess balance. The equipment is a force platform (40 × 60 cm, sampling frequency of 25 Hz) with four implanted strain gauges that determine the center-of-pression (COP) excursion area while the subject stands on it. The BTrackS sampling frequency satisfied the Nyquist theorem for the slow (<10 Hz) COP changes measured in the present study. The force platform has the same accuracy/precision of a laboratory-grade force platform [49]. The force platform was leveled via adjustable legs and verified with a leveling tool, and connected to the computer through a USB cable which also provided power to the platform. The following COP parameters were extracted from the raw data by using the Explore Balance software (Balance Tracking System, San Diego, CA, USA): path length (total sway length in cm); mean velocity (path length divided by trial duration in cm/s); 95% confidence interval ellipse area (the smallest ellipse fitting 95% of COP data in cm^2^) or simply sway area; mediolateral (ML) excursion (the maximal minus the minimal COP data points in medial-lateral direction in cm); anteroposterior (AP) excursion (the maximal minus the minimal COP data points in the anterior-posterior direction in cm). All participants performed three 20 s one-legged stance trials with open eyes and without shoes. The force platform’s manufacturer recommends this amount of time as a standard to ensure reliability. Trials were conducted in a closed room to reduce noise and disturbances. All balance testing was performed by the same experienced researcher. Participants were instructed to look straight ahead at a target placed on a wall at eye level 2 m away, with their arms resting at their side. Repeatability of foot placement between trials was maintained by tapes fixed on top of the force platform. Participants were attached to a belt fixed to the wall to prevent falls during testing and a rater stood close to the participant during all trials.

### 2.3. Dorsiflexion Strength Test

A calibrated hydraulic dynamometer (Saehan Co., Changwon, South Korea) was used for left and right dorsiflexion strength assessments. The patient performed the test three times with the right and left feet, using regular shoes while sitting with arms along the body, and hips and knees flexed at 90° [50]. The maximal ankle dorsiflexion was then asked, and the handle was positioned right below the metatarsophalangeal joint line [50]. The rater applied the resistance against the participant’s dorsiflexion. Participants held the contraction for up to 3 seconds. For further analysis, the average value (in kilograms-force) was calculated from right and left dorsiflexion.

### 2.4. Intervention Protocol

Before and after each session, a stretching exercises routine was performed for 30 s-each (Figure 1).

The RT + P group performed a protocol of resistance exercises (Figure 2) with velocity exercises controlled by the Pilates breathing technique (the first session was used for familiarization purposes with the technique), while the RT group performed the same set of exercises without the Pilates breathing technique. Both groups attended 50-min supervised sessions twice a week for eight weeks. A 2-min rest interval was allowed between each exercise and the load was the same over the weeks. The interventions were paired, with the volume modulated by session duration in minutes. Halter and shin guard loads were the same over the entire protocol. The TR + P group performed the type of exercise cadenced by Pilates breathing, which slows down the exercise execution. The RT + P group performed 1 set of 10 repetitions of each exercise while the RT group performed 2 sets of 10 repetitions. The Pilates breathing technique consists of: (1) inhaling through the nose; (2) deeply exhaling through the mouth, with the lips slightly pursed during the exercise movement. The control group performed a single inhalation during the concentric phase and another expiration while the eccentric phase was performed.

### 2.5. Statistical Analysis

The Shapiro–Wilk and Levene’s tests were used to test the normality and the equality of variances, respectively. The normality and the homogeneity were accepted. Student’s *t*-test was used for participants’ characteristics comparison. A mixed-model approach for intention-to-treat analysis with missing values (mixed analysis of variance [ANOVA] with repeated measures) was used to rate differences within- and between-group. All data was reworked using the Bonferroni–Holm’s post hoc test to avoid multiple comparisons. The significance was set at *p* < 0.05. The standardized differences for the comparisons in all variables were analyzed using Cohen’s d effect size (ES). The magnitude of the ES was qualitatively interpreted using the following thresholds: 0.01–0.19; small: 0.20–0.49; moderate: 0.50–0.79; large: 0.8–1.19; very large: 1.2–1.99; and huge: >2 [51]. All the analysis was performed by a blinded researcher with extensive experience using the JAMOVI software (version 1.1.9, 2020; retrieved from https://www.jamovi.org accessed on 20 February 2020). 

## 3. Results

Eighty-one volunteers were assessed. Thirty-seven did not meet the eligibility criteria and they were excluded from the study. The main reasons for exclusion were neurological diseases and orthopedic disorders that impaired performance of the tests and exercises. Those with impaired cognitive status were also excluded. Forty-four volunteers were then included in the study. They were assigned to the groups by age matching in order to balance the two groups at baseline. The analysis losses are depicted in Figure 3.

Pairing the groups allowed similar overall characteristics at baseline. Some participants did not attend all sessions. Participants in the RT + P group attended 10.6 ± 2.8 sessions, and RT group 10.7 ± 2.6. The groups had similar adherence, 76.4% and 75.7% for RT + P and RT groups, respectively. No minimum number of sessions was established for participants to be included in the analysis. Three participants did not attend for final assessments, two in the RT group and one in the RT + P group. All participants who started treatment were included in the analysis, regardless of whether they completed the protocol. An intention-to-treat analysis was chosen to reduce bias, following the CONSORT recommendation that this analysis is preferred for clinical trials. Participants’ characteristics showed no significant between-group differences at baseline (Table 1).

Descriptive analysis of the international physical activity questionnaire showed that the TR + P group was composed of 59% of active elderly people and 41% were irregularly active individuals. In the TR group, the values were 68% and 32%, active and irregularly active, respectively. Regardless of the groups, no participant was classified as very active or inactive. 

A secondary analysis with the sample composed only of older women is presented in the Section 6.

### 3.1. RT vs. RT + P

The groups were comparable at baseline for dorsiflexion strength (right and left) and most of the balance variables (path length, sway velocity, and excursion AP), but the other balance variables were different at baseline (sway area and excursion ML). No between-group differences were observed for path length (F = 0.05; *p* = 0.81), sway velocity (F = 0.001; *p* = 0.97), excursion ML (F = 2.62; *p* = 0.11) and excursion AP (F = 3.8; *p* = 0.06). However, the RT group showed higher dorsiflexion strength after the protocol compared to the RT + P group (Right: F = 5.05; *p* = 0.031, Left: F = 4.61; *p* = 0.03), with large effect sizes. The RT + P group also showed higher sway area before the protocol compared to the RT group (F = 4.44; *p* = 0.04), with moderate effect size. This difference was not observed after the protocol.

### 3.2. Baseline vs. Post

The RT group showed higher dorsiflexion strength after the protocol compared to baseline (Right: F = 15.75; *p* = 0.001 and Left: F = 24.69; *p* = 0.001), with very large effect sizes. For the RT + P group, all balance variables were lower after the protocol compared to baseline. Moderate effect sizes were observed in pairwise comparisons for path length, sway area, and excursion ML and AP. Large effect size was observed for the sway velocity. Pairwise comparisons are detailed on Table 2.

## 4. Discussion

Balance and strength parameters are indicative of physical impairments and are predictors of future injurious outcomes in older adults [3,9,10,11]. The current study aimed to compare the balance and the dorsiflexion strength of older adults after eight weeks of resistance training with movement velocities controlled by the Pilates breathing technique and the volume modulated by the session duration. The hypothesis was that RT with additional Pilates breathing technique controlling the exercise velocity would prospectively improve balance and muscle strength in older adults. The results showed that only the RT group improved in dorsiflexion strength after the protocol. On the other hand, only the RT + P group improved all balance parameters compared to its own baseline. However, and despite the observed moderate to large effect sizes, the balance improvements were not statistically different from RT group results after the protocol. A possible reason for that is the RT + P worsened balance sway area compared to the RT group on baseline. Thus, the observed moderate effect on the RT + P group normalized the sway area and the other balance parameters to the level of the RT group who begun the protocol with lower baseline values.

Part of the present hypothesis was not fulfilled, as it seems that the number of sets and repetitions may play an important role for dorsiflexion strength, instead of slow movement velocity controlled by the Pilates breathing technique. In the present study, the volume was controlled by the session time (50 min) and not by the number of repetitions and/or sets. This structure allowed the RT group to perform the entire exercise protocol twice, while the RT + P spent more time performing a single set of exercises due to the slower movements’ execution velocity. Other studies showed that RT with slower movement velocity (3 s-concentric phase, 3 s-eccentric phase) and moderate-low load produce greater gains in muscle strength than resistance training with normal velocity in older people [40,41], but some between-studies differences must be highlighted. The open kinetic chain is often the choice for those studies using resistance exercises, while the volume is often controlled by the number of sets and repetitions. Exercises using the body weight as load in closed kinetic chain also keep the velocity of execution constant and the volume equally controlled by sets and repetitions [52]. In both designs and compared to baseline, increased quadriceps muscle strength was found in the experimental group after the protocol with no between-group differences. It seems that the number of sets and repetitions may play a more important role for muscle strength than the movement velocity. In addition, normative values for maximum dorsiflexion strength in older adults range from ~17 to ~28 Kgf [53,54]. Thus, the current baseline values were already considered normal and the final results for the RT group were slightly above the upper limit of dorsiflexion strength.

The current study’s hypothesis was partially fulfilled, considering the improvements in all balance variables in the RT + P group compared to its own baseline. The breathing exercise’s role to improve physical functions has been previously demonstrated. A study using unsupervised home inspiratory muscle training twice a day for eight weeks concluded that the training protocol improved inspiratory muscle function and balance ability in community-dwelling older adults [55]. More recently, the same authors compared the effects of inspiratory muscle training for eight weeks using the well-established OTAGO protocol to improve balance in older adults [56]. Both groups (control and experimental) improved in dynamic balance. In contrast to the last study [50], the present work only assessed the semi-static balance, finding moderate to large effects on balance ability restricted to the group who performed the RT in a breathing-controlled slower manner. Despite the absence of between-group differences right after the intervention protocol, the RT group did not improve in any balance variable even though the exercises were performed twice in each training session. This result disagrees with those found in a recent systematic review that suggested the resistance training as an adequate and effective method to improve balance in older people [57]. However, from the twelve studies included in the review, only one used an exercise modality similar to the present study (body weight and free weights as loads). However, the protocol was longer than the current intervention in weeks. Perhaps longer periods of intervention are necessary to provoke significant improvements in balance when RT is performed. Nevertheless, eight weeks was sufficient to achieve improvements in balance parameters when RT was associated with Pilates breathing.

Aging is commonly accompanied by decreases in muscle mass, strength and power. These impairments are in part attributed to mitochondrial changes and RT is crucial to improve the mitochondrial metabolism dynamics and consequently minimize those impairments. Interestingly, the RT group has shown improved dorsiflexion strength, but not accompanied by any balance improvement though. This result contrasts with a previous study that linked improvements in balance parameters after eight weeks of RT for dorsiflexors [58]. Another previous study showed balance improvements after RT for dorsiflexors compared to a control group also after eight weeks of training [58]. The authors concluded that the improved ankle dorsiflexion strength enhanced the balance performance in older adults. These results are not in agreement with the present findings, as the RT group showed increased dorsiflexion strength but no within-group differences of any balance parameter after the protocol. Another study suggested that the ankle muscle strength is less important than foot sensory function in contributing to postural stability in quiet stance [59], suggesting that the plantar flexor muscle strength would be more associated with fall prevention given its key contribution for reaching tasks in older adults.

The main objective of any intervention to improve balance is the prevention of falls. In this sense, a review with meta-analysis concluded that there is still uncertainty that resistance training alone has favorable effects for the prevention of falls [60]. Our findings are in line with this conclusion, since the RT group showed an increase in dorsiflexion strength with no difference in any balance parameter. On the other hand, the balance improvement results were evident when RT was associated with Pilates breathing technique to modulate the velocity exercise. The results suggest that the variables of exercise (execution speed, number of sets, repetitions, and total session duration) may be modulated differently in RT when the goals are not the same (to improve strength or balance).

Some limitations must be addressed in the present study. Firstly, the sample was not randomized due to age-matched pairing. The matching was required to ensure homogeneity of the participants’ characteristics at baseline. The sample mainly constituted of older women. The authors opted to recruit community-dwelling older adults under clinical care due to convenience, and women are often more prone to attend for physical care than men. Thus, the current findings may change with more gender-balanced samples. Finally, some participants did not attend for final assessments, adherence was not fully fulfilled and some participants missed some training sessions, which is expected in clinical trials. The lack of final assessments and the faulty attendance could impair exact interpretation, although it was similar between groups. Nevertheless, the intention-to-treat analysis kept all participants initially included in the study to fit the statistical evaluation to the study design.

## 5. Conclusions

The results suggest that eight weeks of RT performing doubled sets of exercise improved dorsiflexion strength in a between-group comparison, without any effect in static balance. The experimental protocol of RT with slow velocity movement cadence by the Pilates breathing technique allowed prospective balance improvements, compared to within-group baseline assessments. Thus, the slow velocity of movement cadence by Pilates breathing technique influenced the static balance. These results suggest that RT varies as the movement velocity, the number of sets, repetitions and the total session duration might vary according to the objective (strength and balance enhancement). 

## 6. Supplementary Section

As the sample consisted mostly of older women, the few included older men could influence the results. Therefore, a secondary analysis was performed with the sample composed only of older women. 

Participants’ characteristics showed no significant between-group differences at baseline (Table 3).

Descriptive values of handgrip strength remained similar to the first analysis, despite the sample being composed only of older women. 

The results for dorsiflexor strength and balance are presented below (Table 4).

Dorsiflexion strength results were very similar and did not impact the discussion nor the conclusions. The RT group showed higher right and left dorsiflexion strength after the protocol compared to baseline with very large and large effect sizes, respectively. The RT + P group did not improve dorsiflexion strength (Table 4). 

Despite the improvement in dorsiflexors strength, the RT group did not improve balance. On the other hand, all balance variables were lower after the protocol compared to baseline in the RT + P group, with large effect sizes for path length and sway velocity. Moderate effect sizes were observed in pairwise comparisons for sway area and excursion ML and AP.

Despite the very large, large, and moderate effect sizes observed in the improvement of balance in the RT + P group, no statistical difference was observed in the balance between the groups after the protocol. 

Between-group pairwise comparisons differences at baseline in sway area and excursion ML were observed. It is possible that the improvement in the RT + P group normalized these CoP oscillation variables.

The present secondary analysis leads to the same conclusions. RT performing doubled sets of exercising improved dorsiflexion strength without any significance in static balance. The RT + P group with slow movement cadenced by the Pilates breathing technique allowed prospective balance improvements.

## Figures and Tables

**Figure 1 ijerph-19-10849-f001:**
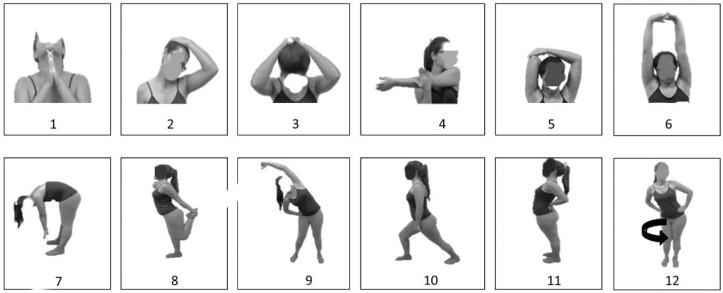
Stretching protocol. (**1**). Head extension; (**2**). Head lateroflexion; (**3**). Head flexion; (**4**). Shoulders; (**5**). Triceps; (**6**). Arm stretch; (**7**). Hamstrings; (**8**). Quadriceps; (**9**). Trunk lateroflexion; (**10**). Gastrocnemius; (**11**). Trunk extension; (**12**). Hip circle.

**Figure 2 ijerph-19-10849-f002:**
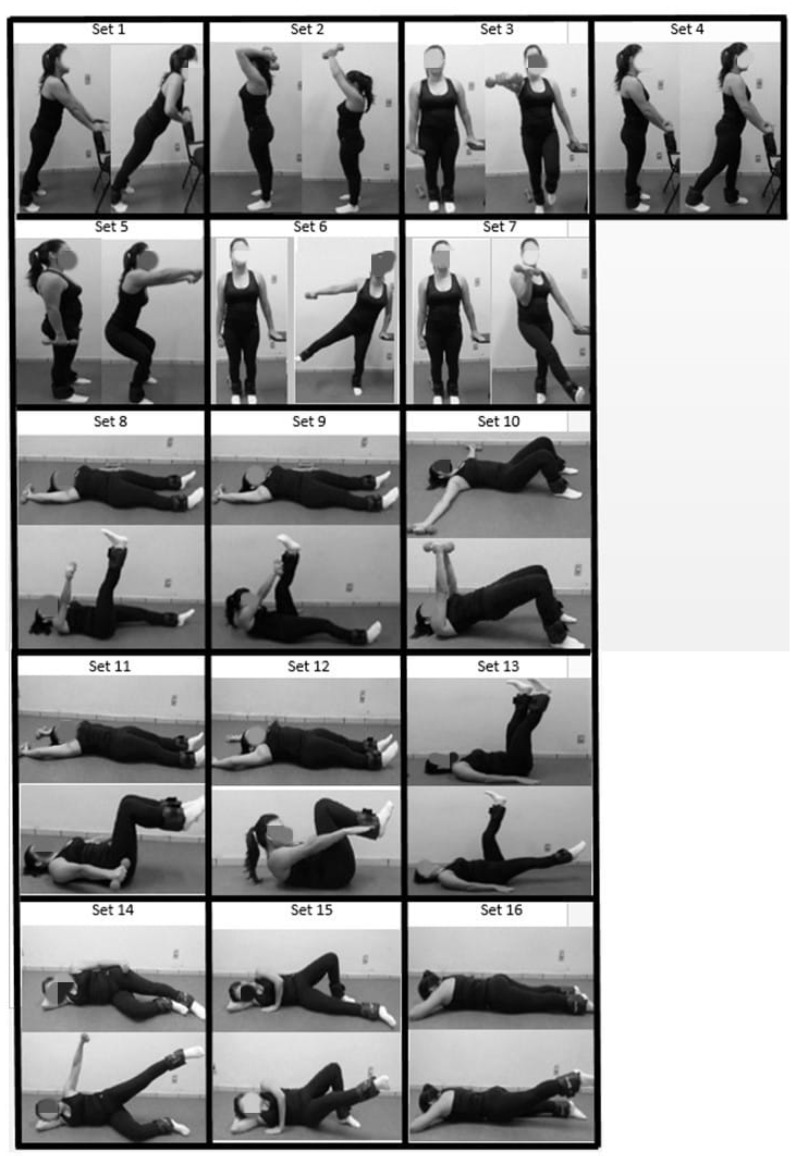
Exercise protocol. Each set was composed of 10 repetitions of each exercise. Halter and shin guard loads were the same over the entire protocol. Set 1. In supine: hip flexion (with a shin guard load) + ipsilateral shoulder extension holding the halter; Set 2. In supine: hip flexion (with a shin guard load) + contralateral shoulder extension holding the halter; Set 3. In supine: hip elevation (bridge) + shoulder horizontal adduction (with halters); Set 4. In supine: hip and knee flexion (with shin guard loads) + shoulder extension (with halters); Set 5. In supine: hip and knee flexion (with shin guard loads) + shoulder extension + trunk elevation (curl-up); Set 6. In both lateral decubitus: hip and shoulder abduction using halter and shin guard loads; Set 7. In both lateral decubitus: hip adduction with shin guard load; Set 8. In prone: hip extension (~10 degrees) in both limbs with shin guard load; Set 9. In supine: hip flexion (~90 degrees) alternating the lower limbs with shin guard loads; Set 10. Standing push-ups with hands on a chair; Set 11. Triceps strengthening, elbow flexion-extension with both limbs; Set 12. Standing with hands on a chair: hip extension with shin guard loads alternating both limbs; Set 13. Squatting + shoulder flexion (~90 degrees); Set 14. Standing with hands on a chair: single-leg plantar flexion + shoulder flexion (90 degrees with halter) for both sides; Set 15. Standing hip (with shin guard load) + shoulder abduction (with halter) for both sides; Set 16. Standing hip adduction (with shin guard load) + elbow flexion (with halter) for both sides.

**Figure 3 ijerph-19-10849-f003:**
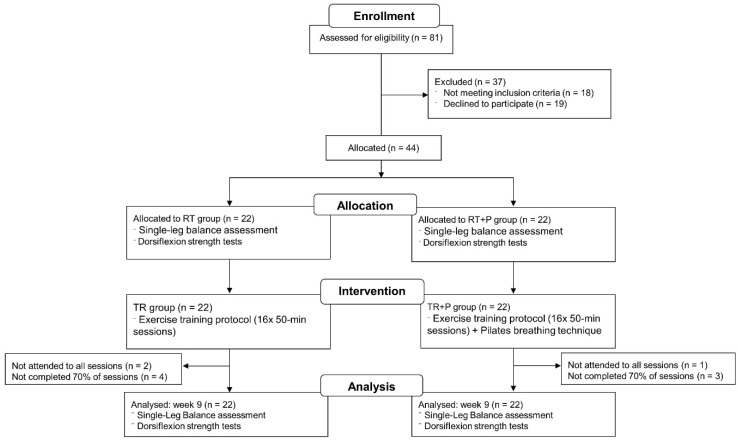
Flow chart. RT: resistance training; RT + P: resistance training with Pilates breathing technique.

**Table 1 ijerph-19-10849-t001:** Participants’ characteristics.

Outcome	Groups		*p*-Value
	RT	RT + P	
**n (%male/%female)**	22 (14/86)	22 (14/86)	
Age (years)	70 ± 6	69 ± 6	0.567
BMI (Kg/m^2^)	25.4 ± 5.0	26.1 ± 4.5	0.580
Diabetes (%yes/%no)	1 (5/95)	2 (9/91)	0.305
FES-I	20.0 ± 2.4	21.3 ± 4.3	0.217
R-Handgrip (Kgf)	23.9 ± 5.8	23.4 ± 8.0	0.830
Falls History (%yes/%no)	4 (18/82)	4 (18/82)	0.689
MiniMental	25.8 ± 2.7	25.8 ± 2.1	0.973
SBP (mmHg)	122.0 ± 13.4	125.0 ± 16.6	0.510
DBP (mmHg)	73.5 ± 9.7	77.3 ± 10.0	0.221

Data are presented as mean ± standard deviation ou N (%); BMI = Body mass index; FES-I = Falls Efficacy Scale; R = right; MiniMental = Mini Mental State Examination; SBP = systolic blood pressure; DBP = diastolic blood pressure; RT: resistance training; RT + P: resistance training with Pilates breathing technique.

**Table 2 ijerph-19-10849-t002:** Between-group and within-group pairwise comparisons.

Outcome	RT Group		RT + P Group		Between-Group Pairwise Comparisons (*p*-Value [ES])	Within-Group Pairwise Comparisons (*p*-Value [ES])
	Baseline (1)	Post (2)	Baseline (3)	Post (4)		
R-Dorsiflexion (Kgf)	22.2 ± 4.2	29.1 ± 7.7	21.2 ± 7.7	22.9 ± 5.2	1;3;4 < 2 (0.001 [0.96])	1 < 2 (0.002 [1.09—very large])
L-Dorsiflexion (Kgf)	22.4 ± 3.3	29.5 ± 6.9	21.9 ± 7.3	24.0 ± 5.2	1;3;4 < 2 (0.001 [0.92])	1 < 2 (0.001 [1.06—very large])
Path Length (cm)	70.8 ± 15.7	63.1 ± 11.7	71.0 ± 14.3	59.7 ± 14.3	NS	3 > 4 (0.003 [0.87—large])
Sway Velocity (cm/s)	3.5 ± 0.8	3.1 ± 0.6	3.6 ± 0.7	2.9 ± 0.7	NS	3 > 4 (0.001 [0.90—large])
Sway Area (cm^2^)	6.6 ± 2.9	5.3 ± 1.7	8.9 ± 5.3	5.7 ± 2.1	3 > 1 (0.01 [0.52])	3 > 4 (0.003 [0.71—moderate])
Excursion ML (cm)	2.6 ± 0.5	2.6 ± 0.3	3.0 ± 0.7	2.6 ± 0.5	NS	3 > 4 (0.002 [0.77—moderate])
Excursion AP (cm)	3.1 ± 0.9	2.6 ± 0.6	3.6 ± 1.4	2.8 ± 0.7	NS	3 > 4 (0.010 [0.63—moderate)

Data are presented as mean ± standard deviation; R = right; L = Left; ML = mediolateral; AP = anteroposterior; ES = Effect size; NS = non-significant; RT: resistance training; RT + P: resistance training with Pilates breathing technique.

**Table 3 ijerph-19-10849-t003:** Participants’ characteristics.

Outcome	Groups		*p*-Value
	RT	RT + P	
n	19	19	
Age (years)	70.5 ± 6.19	68.2 ± 5.9	0.246
BMI (Kg/m^2^)	24.8 ± 4.8	26.1 ± 4.8	0.413
Diabetes (%yes/%no)	1 (5/95)	3 (16/84)	0.305
FES-I	20.7 ± 4.4	20.3 ± 2.4	0.680
R-Handgrip (Kgf)	22.3 ± 3.5	21.6 ± 6	0.662
Falls History (%yes/%no)	4 (21/79)	2 (11/89)	0.689
MiniMental	25.5 ± 2.7	25.6 ± 2.1	0.944
SBP (mmHg)	122.0 ± 13.6	125.0 ± 17.8	0.591
DBP (mmHg)	73.6 ± 10.1	77.8 ± 10.6	0.218

Data are presented as mean ± standard deviation ou N (%); BMI = Body mass index; FES-I = Falls Efficacy Scale; R = right; MiniMental = Mini Mental State Examination; SBP = systolic blood pressure; DBP = diastolic blood pressure; RT: resistance training; RT + P: resistance training with Pilates breathing technique.

**Table 4 ijerph-19-10849-t004:** Between-group and within-group pairwise comparisons.

Outcome	RT Group		RT + P Group		Between-Group Pairwise Comparisons (*p*-Value [ES])	Within-Group Pairwise Comparisons (*p*-Value [ES])
	Baseline (1)	Post (2)	Baseline (3)	Post (4)		
R-Dorsiflexion (Kgf)	21.4 ± 3.6	27.8 ± 7.2	19.5 ± 6.2	21.9 ± 4.6	1;3;4 < 2 (0.002 [0.97])	1 < 2 (0.002 [1.02—large])
L-Dorsiflexion (Kgf)	21.8 ± 2.9	28.1 ± 6.0	20.0 ± 6.04	22.7 ± 4.2	1;3;4 < 2 (0.001 [1.04])	1 < 2 (0.001 [1.21—very large])
Path Length (cm)	71.0 ± 16.9	63.9 ± 12.2	69.6 ± 14.8	56.8 ± 12.1	NS	3 > 4 (0.003 [0.94—large])
Sway Velocity (cm/s)	3.55 ± 0.8	3.19 ± 0.6	3.6 ± 0.7	2.8 ± 0.6	NS	3 > 4 (0.001 [1.21—large])
Sway Area (cm^2^)	6.50 ± 3.0	5.17 ± 1.8	9.2 ± 6.5	5.7 ± 5.2	3 > 1 (0.015 [0.53])	3 > 4 (0.003 [0.58—moderate])
Excursion ML (cm)	2.62 ± 0.5	2.57 ± 0.3	3.0 ± 0.7	2.6 ± 0.5	3 > 1 (0.020 [0.67])	3 > 4 (0.002 [0.64—moderate])
Excursion AP (cm)	3.08 ± 1.0	2.64 ± 0.7	3.7 ± 1.5	2.8 ± 0.7	NS	3 > 4 (0.010 [0.68—moderate])

Data are presented as mean ± standard deviation; R = right; L = Left; ML = mediolateral; AP = anteroposterior; ES = effect size; NS = non-significant; RT: resistance training; RT + P: resistance training with Pilates breathing technique.

## Data Availability

Data are available only upon request to the authors.

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
