# Peer review of "Does 8-Week Resistance Training with Slow Movement Cadenced by Pilates Breathing Affect Muscle Strength and Balance of Older Adults? An Age-Matched Controlled Trial"

_ijerph, 2022, doi:10.3390/ijerph191710849_

Round 1
Reviewer 1 Report
Dear Authors,
In my opinion, this topic is very interesting, however, I have some doubts regarding the study methodology and other critical issues that have to be clarified to improve the paper.
Major revisions:
INTRODUCTION: This section should be improved by emphasizing the need for effective prevention programs to reduce risk of falls and fragility fractures, given their detrimental consequences in older adults.
According to this, you should cite the following references:
- Concin H, Brozek W, Benedetto KP, Häfele H, Kopf J, Bärenzung T, Schnetzer R, Schenk C, Stimpfl E, Waheed-Hutter U, Ulmer H, Rapp K, Zwettler E, Nagel G. Hip fracture incidence 2003-2013 and projected cases until 2050 in Austria: a population-based study. Int J Public Health. 2016 Dec;61(9):1021-1030. doi: 10.1007/s00038-016-0878-9.
- de Sire A, Invernizzi M, Baricich A, Lippi L, Ammendolia A, Grassi FA, Leigheb M. Optimization of transdisciplinary management of elderly with femur proximal extremity fracture: A patient-tailored plan from orthopaedics to rehabilitation. World J Orthop. 2021 Jul 18;12(7):456-466. doi: 10.5312/wjo.v12.i7.456.
- Richard L, Gauvin L, Gosselin C, Ducharme F, Sapinski JP, Trudel M. Integrating the ecological approach in health promotion for older adults: a survey of programs aimed at elder abuse prevention, falls prevention, and appropriate medication use. Int J Public Health. 2008;53(1):46-56. doi: 10.1007/s00038-007-6099-5.
MATERIALS AND METHODS: The study design should be characterized.
MATERIALS AND METHODS: This section should be improved by clarifying setting locations, and relevant dates, including periods of recruitment, exposure, follow-up, and data collection of the study.
MATERIALS AND METHODS: The protocol of resistance exercises should be better characterized. The exercise program should be better presented, clarifying time under tension in control group, and supervision.
MATERIALS AND METHODS: It should be clarified who performed the analysis (qualification, degree, and especially if blinded).
MATERIALS AND METHODS: Primary and secondary outcomes should be clarified.
RESULTS: This section should be improved, reporting the numbers of patients assessed for eligibility and the number of patients excluded, clarifying at least the main cause of exclusions.
RESULTS: The sample characterization should be reported. In particular, baseline characteristics should be implemented (i.e. physical activity levels, medications)
RESULTS: The sample size calculation must be described in the statistical methods.
DISCUSSION: This section should be improved by emphasizing the clinical implication of the present article and underlining the role of resistance exercise in targeting the multilevel pathways promoting aging process.
According to this, you should cite the following references:
- Keating CJ, Cabrera-Linares JC, Párraga-Montilla JA, Latorre-Román PA, Del Castillo RM, García-Pinillos F. Influence of Resistance Training on Gait & Balance Parameters in Older Adults: A Systematic Review. Int J Environ Res Public Health. 2021 Feb 11;18(4):1759. doi: 10.3390/ijerph18041759.
- Sadjapong U, Yodkeeree S, Sungkarat S, Siviroj P. Multicomponent Exercise Program Reduces Frailty and Inflammatory Biomarkers and Improves Physical Performance in Community-Dwelling Older Adults: A Randomized Controlled Trial. Int J Environ Res Public Health. 2020 May 26;17(11):3760. doi: 10.3390/ijerph17113760.
- Lippi L, de Sire A, Mezian K, Curci C, Perrero L, Turco A, Andaloro S, Ammendolia A, Fusco N, Invernizzi M. Impact of exercise training on muscle mitochondria modifications in older adults: a systematic review of randomized controlled trials. Aging Clin Exp Res. 2022 Jul;34(7):1495-1510. doi: 10.1007/s40520-021-02073-w.
Minor revisions:
ABSTRACT: Line 25. Typing error. Please replace “ende” with “end”
KEYWORDS. I suggest including “training”, and “breathing” between keywords.
Author Response
REVIEW 1
Comments and Suggestions for Authors
Dear Authors,
In my opinion, this topic is very interesting, however, I have some doubts regarding the study methodology and other critical issues that have to be clarified to improve the paper.
Major revisions:
INTRODUCTION: This section should be improved by emphasizing the need for effective prevention programs to reduce risk of falls and fragility fractures, given their detrimental consequences in older adults.
According to this, you should cite the following references:
- Concin H, Brozek W, Benedetto KP, Häfele H, Kopf J, Bärenzung T, Schnetzer R, Schenk C, Stimpfl E, Waheed-Hutter U, Ulmer H, Rapp K, Zwettler E, Nagel G. Hip fracture incidence 2003-2013 and projected cases until 2050 in Austria: a population-based study. Int J Public Health. 2016 Dec;61(9):1021-1030. doi: 10.1007/s00038-016-0878-9.
- de Sire A, Invernizzi M, Baricich A, Lippi L, Ammendolia A, Grassi FA, Leigheb M. Optimization of transdisciplinary management of elderly with femur proximal extremity fracture: A patient-tailored plan from orthopaedics to rehabilitation. World J Orthop. 2021 Jul 18;12(7):456-466. doi: 10.5312/wjo.v12.i7.456.
- Richard L, Gauvin L, Gosselin C, Ducharme F, Sapinski JP, Trudel M. Integrating the ecological approach in health promotion for older adults: a survey of programs aimed at elder abuse prevention, falls prevention, and appropriate medication use. Int J Public Health. 2008;53(1):46-56. doi: 10.1007/s00038-007-6099-5.
ANSWER: Thank you for your comment. You may see our corrected version with all the above mentioned references.
MATERIALS AND METHODS: The study design should be characterized.
ANSWER: Thank you for your attention. We mentioned the study’s design in the corrected version.
MATERIALS AND METHODS: This section should be improved by clarifying setting locations, and relevant dates, including periods of recruitment, exposure, follow-up, and data collection of the study.
ANSWER: Corrected.
MATERIALS AND METHODS: The protocol of resistance exercises should be better characterized. The exercise program should be better presented, clarifying time under tension in control group, and supervision.
ANSWER: Thank you for your attention. It is true that the TR+P group performed the exercise slowly compared to the RT group. However, the execution time of the concentric and eccentric phases was not calculated. Instead, the authors opt to use the breathing as a marker. Thus, the control group performed inspiration during the concentric phase and expiration while the eccentric phase was performed. We added the information in our current version.
MATERIALS AND METHODS: It should be clarified who performed the analysis (qualification, degree, and especially if blinded).
ANSWER: Corrected. Thank you for your comment.
MATERIALS AND METHODS: Primary and secondary outcomes should be clarified.
RESULTS: This section should be improved, reporting the numbers of patients assessed for eligibility and the number of patients excluded, clarifying at least the main cause of exclusions.
ANSWER: Thank you for your attention. Corrected.
RESULTS: The sample characterization should be reported. In particular, baseline characteristics should be implemented (i.e. physical activity levels, medications)
ANSWER: The level of physical activity was assessed using the IPAQ questionnaire, but medications in use were not collected.
RESULTS: The sample size calculation must be described in the statistical methods.
ANSWER: Thank you for your attention. However, the calculation was strategically positioned in the participants’ section to provide more flow to our text. To split the analysis would impair the readiness, as you may notice. The exclusion and the group division is also in the same paragraph. We kept the format.
DISCUSSION: This section should be improved by emphasizing the clinical implication of the present article and underlining the role of resistance exercise in targeting the multilevel pathways promoting aging process.
ANSWER: Thank you for your comment. Please, see our current version. We tried to improve the text accord to your guidance.
According to this, you should cite the following references:
- Keating CJ, Cabrera-Linares JC, Párraga-Montilla JA, Latorre-Román PA, Del Castillo RM, García-Pinillos F. Influence of Resistance Training on Gait & Balance Parameters in Older Adults: A Systematic Review. Int J Environ Res Public Health. 2021 Feb 11;18(4):1759. doi: 10.3390/ijerph18041759.
- Sadjapong U, Yodkeeree S, Sungkarat S, Siviroj P. Multicomponent Exercise Program Reduces Frailty and Inflammatory Biomarkers and Improves Physical Performance in Community-Dwelling Older Adults: A Randomized Controlled Trial. Int J Environ Res Public Health. 2020 May 26;17(11):3760. doi: 10.3390/ijerph17113760.
- Lippi L, de Sire A, Mezian K, Curci C, Perrero L, Turco A, Andaloro S, Ammendolia A, Fusco N, Invernizzi M. Impact of exercise training on muscle mitochondria modifications in older adults: a systematic review of randomized controlled trials. Aging Clin Exp Res. 2022 Jul;34(7):1495-1510. doi: 10.1007/s40520-021-02073-w.
ANSWER: Thank you for your comment. We inserted 2 of 3. Please, see the corrected version. One reference did not fit to our current rationale.
Minor revisions:
ABSTRACT: Line 25. Typing error. Please replace “ende” with “end”
KEYWORDS. I suggest including “training”, and “breathing” between keywords.
ANSWER: Corrected. Thank you for your attention.
Reviewer 2 Report
The authors have conducted a study to test the combined effects of RT and pilates in older adults. The findings are as expected. There are few comments below for the authors to consider.
Abstract: "The improvements on balance were not statistically different from RT group." this sentence leads no where.
Please consider providing some numbers and P-values in the abstract. Only text is not sufficient.
Introduction: Meriam Nelson and Steve Ball have conducted a lot of studies in older adults with RT programs. The authors are recommended to consider reading their work and include the citations in the introduction and discussion as applicable. Examples below.
Morganti, C. M., Nelson, M. E., Fiatarone, M. A., Dallal, G. E., Economos, C. D., Crawford, B. M., & Evans, W. J. (1995). Strength improvements with 1 yr of progressive resistance training in older women. Medicine and science in sports and exercise, 27(6), 906-912.
Syed-Abdul, M. M., McClellan, C. L., Parks, E. J., & Ball, S. D. (2022). Effects of a resistance training community programme in older adults. Ageing & Society, 42(8), 1863-1878.
The authors may choose to use other articles as well from these authors that fits their intro and discussion.
Methods: Images are not clear, hope the final version will include clear images.
Please add the ethics board approval number, name and address. Was this study registered in ClinicalTrials.gov? If yes, please provide the CT number as well.
Please provide a clear picture of the consort flow diagram. It's difficult to read for some individuals.
Results: If 6 participants in RT and 4 in RT+P groups did not attend all sections, why did the authors decided to use their data in the analysis? if this is true, please explain. If not, please correct the information in the fig 3.
Discussion:
Since there were more women, how does the results change when only women are included in the analysis? This secondary analysis can be provided in the supplementary section.
Other comments:
The authors need to provide the information related to adherence, how many sessions were attended/missed by participants, and if there was any requirement for participant to attend certain number of sessions to be included in the analysis?
Author Response
REVIEW 2
Comments and Suggestions for Authors
The authors have conducted a study to test the combined effects of RT and pilates in older adults. The findings are as expected. There are few comments below for the authors to consider.
ABSTRACT: "The improvements on balance were not statistically different from RT group." this sentence leads no where. Please consider providing some numbers and P-values in the abstract. Only text is not sufficient.
ANSWER: Thank you for your comment. Corrected accordingly.
INTRODUCTION: Meriam Nelson and Steve Ball have conducted a lot of studies in older adults with RT programs. The authors are recommended to consider reading their work and include the citations in the introduction and discussion as applicable. Examples below.
Morganti, C. M., Nelson, M. E., Fiatarone, M. A., Dallal, G. E., Economos, C. D., Crawford, B. M., & Evans, W. J. (1995). Strength improvements with 1 yr of progressive resistance training in older women. Medicine and science in sports and exercise, 27(6), 906-912.
Syed-Abdul, M. M., McClellan, C. L., Parks, E. J., & Ball, S. D. (2022). Effects of a resistance training community programme in older adults. Ageing & Society, 42(8), 1863-1878.
ANSWER: Thank you for your assistance. We added the references.
The authors may choose to use other articles as well from these authors that fits their intro and discussion.
METHODS: Images are not clear, hope the final version will include clear images.
ANSWER: Corrected.
Please add the ethics board approval number, name and address. Was this study registered in ClinicalTrials.gov? If yes, please provide the CT number as well.
ANSWER: Corrected.
Please provide a clear picture of the consort flow diagram. It's difficult to read for some individuals.
ANSWER: Corrected.
RESULTS: If 6 participants in RT and 4 in RT+P groups did not attend all sections, why did the authors decided to use their data in the analysis? if this is true, please explain. If not, please correct the information in the fig 3.
ANSWER: Although the sample was matched, we obtained a sample with similar general characteristics at the beginning of the treatment, so an intention-to-treat analysis was chosen in order to reduce bias, following the Consort recommendation, that this is the preferential analysis for clinical trials.
DISCUSSION:
Since there were more women, how does the results change when only women are included in the analysis? This secondary analysis can be provided in the supplementary section.
ANSWER: Thank you for your comment. We added the secondary analysis. Please, see our corrected version.
OTHER COMMENTS:
The authors need to provide the information related to adherence, how many sessions were attended/missed by participants, and if there was any requirement for participant to attend certain number of sessions to be included in the analysis?
ANSWER: The info was added to our current version. Thank you for your assistance.
Reviewer 3 Report
General comments
The authors compared the balance and the dorsiflexion strength of older adults after 8-week of resistance training with movement velocities controlled by the Pilates breathing technique and the volume modulated by the session duration.
I really like this study. It is very well written. I have only minor recommendations for authors.
Specific comments
Line 4: Replace with "An" (capital A).
Line 25: Replace with "at the end of the protocol...".
Introduction
The authors have done a good job of synthesizing the literature. They clearly tell what gaps in the literature they are trying to fill in.
Materials and Methods
The methodology is clearly explained.
The instruments used are validated and reliable.
Statistics are appropriate.
Results
Figure 3: Please replace the figure with a more readable one.
Discussion
The authors' conclusions are justified.
The writing is clear and to the point.
The limitations were addressed.
Conclusions
The take-home message is clear.
What direction should research work take in this area? Add future directions/implications.
Line 334: Replace “de” with “the”.
Author Response
REVIEW 3
General comments
The authors compared the balance and the dorsiflexion strength of older adults after 8-week of resistance training with movement velocities controlled by the Pilates breathing technique and the volume modulated by the session duration.
I really like this study. It is very well written. I have only minor recommendations for authors.
Specific comments
Line 4: Replace with "An" (capital A).
Line 25: Replace with "at the end of the protocol...".
ANSWER: Corrected.
Introduction
The authors have done a good job of synthesizing the literature. They clearly tell what gaps in the literature they are trying to fill in.
Materials and Methods
The methodology is clearly explained.
The instruments used are validated and reliable.
Statistics are appropriate.
Results
Figure 3: Please replace the figure with a more readable one.
ANSWER: Corrected.
Discussion
The authors' conclusions are justified.
The writing is clear and to the point.
The limitations were addressed.
Conclusions
The take-home message is clear.
What direction should research work take in this area? Add future directions/implications.
It is highlighted in text.
Line 334: Replace “de” with “the”.
ANSWER: Corrected.
Round 2
Reviewer 1 Report
Dear Authors,
in my opinion, the manuscript is interesting, and the results are intriguing.
You have significantly improved the paper during the revision process.
Therefore, in my opinion, the paper is now suitable for publication in this Journal.
Best regards
Author Response
Dear Authors,
in my opinion, the manuscript is interesting, and the results are intriguing.
You have significantly improved the paper during the revision process.
Therefore, in my opinion, the paper is now suitable for publication in this Journal.
Best regards
ANSWER: Thank you for your kind comment. We also appreciate your effort to improve our manuscript's clarity.
Reviewer 2 Report
The authors have significantly improved the manuscript and have responded to most of my comments. However, I believe that the authors should seriously consider including participants who attended at least 60% of the sessions. This is critical for the conclusions presented here. Without the attendance of the participants, the effect cannot be truly related to the intervention.
Ln 45, 67 - typo in references
Ln 196 - typo
Ln 260-264 - I understand that the authors were trying to conduct intention to treat analysis. However, if a participant completed only 1 session, was the effect true and can be related to RT or RT+P? I don't think so. I think the authors should consider including participants in the analysis those who completed at least 60% of the sessions. Otherwise, it's unclear if these effects were due to intervention or not. These are only my suggestions and authors are recommended to consider them. If the authors do not wish to exclude the participants, it is their discretion. However, they must clearly provide the exact number of sessions each participant attended or the mean±SD of each group. The authors vaguely mentions on Ln 375 that the adherence was not fulfilled but this is not clear as to how many sessions each participant attended or does not show the exact number of sessions each group attended. This information must be provided for the readers to understand that the effects observed are results of how many sessions on an average.
Ln 376 - "some participants did not attend final assessments" what does this mean? Did the participants not show up for their follow-up visit? if yes, then how can the authors claim to have 22 participants for follow-up analysis? If n=22 then put the actual n in the table 2 and this number must also be reflected in fig 3. The authors should seriously consider what was their n and must specify clearly in fig 3, table 1, table 2,
Consider adding a discussion of the supplementary secondary analysis. These must also be mentioned briefly in the results section.
Author Response
Comments and Suggestions for Authors
The authors have significantly improved the manuscript and have responded to most of my comments. However, I believe that the authors should seriously consider including participants who attended at least 60% of the sessions. This is critical for the conclusions presented here. Without the attendance of the participants, the effect cannot be truly related to the intervention.
Ln 45, 67 - typo in references
ANSWER: Corrected.
Ln 196 – typo
ANSWER:Could you mark it, please? We did not find the typo.
Ln 260-264 - I understand that the authors were trying to conduct intention to treat analysis. However, if a participant completed only 1 session, was the effect true and can be related to RT or RT+P? I don't think so. I think the authors should consider including participants in the analysis those who completed at least 60% of the sessions. Otherwise, it's unclear if these effects were due to intervention or not. These are only my suggestions and authors are recommended to consider them. If the authors do not wish to exclude the participants, it is their discretion. However, they must clearly provide the exact number of sessions each participant attended or the mean±SD of each group. The authors vaguely mentions on Ln 375 that the adherence was not fulfilled but this is not clear as to how many sessions each participant attended or does not show the exact number of sessions each group attended. This information must be provided for the readers to understand that the effects observed are results of how many sessions on an average.
ANSWER: Thank you for your comment. Our choice for intention-to-treat analysis follows the CONSORT recommendation. The groups had similar adherence, 76.4% and 75.7% for RT+P and RT groups respectively. Participants in the RT+P group attended 10.6 ± 2.8 sessions, and RT group 10.7± 2.6.
Ln 376 - "some participants did not attend final assessments" what does this mean? Did the participants not show up for their follow-up visit? if yes, then how can the authors claim to have 22 participants for follow-up analysis? If n=22 then put the actual n in the table 2 and this number must also be reflected in fig 3. The authors should seriously consider what was their n and must specify clearly in fig 3, table 1, table 2,
ANSWER: Thank you for your comment. Please, see our corrected manuscript. We added an explanation.
Consider adding a discussion of the supplementary secondary analysis.
ANSWER: The secondary analysis was added.
These must also be mentioned briefly in the results section.
ANSWER: Thank you for your attention. Corrected.